# An Open Trial on the Feasibility and Efficacy of a Mindfulness-Based Intervention with Psychoeducational Elements on Atopic Eczema and Chronic Itch

**Julia Harfensteller**

SAGE Institute for Mindfulness and Health, 10115 Berlin, Germany; harfensteller@sage-institut.de

**Abstract:** (1) Background: Mindfulness-based interventions (MBI) are psychological group interventions conducted over several weeks. Their effects on reducing stress and improving physical and psychological health have been proven in various clinical populations. Growing evidence suggests that MBIs might be beneficial for dermatology patients. This article reports on a novel Mindfulness-based Training for chronic Skin Conditions (MBTSC) with psychoeducational elements that was developed with the goal of improving self-regulation including stress management and emotion regulation in patients and to help in coping with disease symptoms such as itch and scratching. The intervention was tested in a pilot efficacy trial in order to examine feasibility and to collect preliminary data on the effectiveness of the intervention on disease severity including itch perception and on psychological distress in an atopic dermatitis (AD) sample. (2) Methods: Following an uncontrolled pre-test-post-test design based on standardized self-report measures, nine adult AD patients were recruited from a dermatology clinic. Data were collected at baseline, post-treatment and 3 month follow-up. Patients completed questionnaires assessing disease severity, itch perception, stress, anxiety and depression, mindfulness and intervention acceptability. The 7 week intervention included seven weekly sessions and a daily home-practice requirement, supported by guided audio-meditations and reading material. (3) Results: Quantitative data showed improvements in disease severity, itch perception and stress levels with small to medium effect sizes. Psychological distress increased at post-treatment—significantly in the case of depression. Qualitative data highlighted the mixed effects of MBTSC on symptoms. Treatment acceptability was high and 100% of the participants completed the intervention; (4) Conclusions: These data indicate that MBTSC is feasible and that it might be a useful tool as adjunct therapy for AD. Further studies with larger samples and control groups are needed.

**Keywords:** mindfulness; atopic eczema; dermatitis; skin; itch; pruritus; psychodermatology; stress regulation; emotion; emotion regulation



## 1. Introduction

Atopic dermatitis (AD) is a chronically relapsing inflammatory skin disease that is characterized by itching and eczematous skin lesions due to scratching. Itching plays a key role in the exacerbation and maintenance of AD because it causes scratching and also because it is a highly distressing experience in itself [1]. The vicious itch–scratch circle together with stigmatizing skin distortion on account of eczema brings about serious psychosocial distress, and significantly reduces quality of life [2]. The effects of psychoemotional factors such as stress on AD are widely accepted but only in recent years serious advances have been made in uncovering the complex relationship between stress and AD. As a major trigger factor for AD flares, stress has been shown to be associated with (i) an impaired skin-barrier function, (ii) increased itch perception and (iv) scratching behavior [3–5]. In addition to stress, emotions and emotion regulation are considered serious vulnerability factors in psychosomatic dynamics of skin conditions [6]. Growing

empirical evidence shows that negative emotions, and more importantly, dysfunctional emotion regulation, such as unrecognized anger and aversion, may have a negative effect on symptoms and may modulate itch sensitivity [7–9]. With regard to emotion regulation, particularly poor emotional awareness and difficulties to "read" emotions (alexithymia) have been widely documented to be associated with skin conditions [10]. For instance, a controlled study showed that AD patients have considerably higher rates of alexithymia than skin-healthy controls [11]. Although pathogenic mechanisms are not quite clear, it is assumed that the lack of emotional awareness might lead to physiological hyperarousal and somatization processes might play a moderating role in the emotion–skin as well as in the emotion–itch relationship [12,13].

On the backdrop of these and other findings, the necessity for interventions targeting stress and stress-related factors such as emotion regulation has repeatedly been emphasized in research [14–16]. In recent decades, mindfulness-based interventions (MBI) have gained a reputation as effective measures for stress reduction in diverse clinical populations, including chronic inflammatory conditions [17,18]. MBIs are behavioral group interventions conducted over several weeks that aim at teaching several cognitive and emotional skills. These mindfulness competences involve (i) non-judgmental present-moment awareness of bodily sensations, cognitions, impulses, feelings, or external stimuli such as sounds or tastes; (ii) attention regulation in the sense of disengaging from (stressful) thoughts and refocusing attention to the present moment; (iii) attitudinal aspects including (self-)acceptance, curiosity and openness [19,20]. Although MBIs employ mindfulness meditation as core practice, they also include other components such as body movement, yoga and psychoeducation. In recent years, a series of adapted interventions has emerged from the original Mindfulness-Based Stress Reduction (MBSR) protocol, in particular because of the need to customize to the specific requirements of clinical populations and settings. These second-generation MBIs such as Mindfulness-based Cognitive Therapy (MBCT) have been proven to be effective as well [21].

In dermatology, an increasing number of studies report that mindfulness is likely to have positive effects on skin conditions [22,23]. Early studies showed that meditation helps to improve symptoms (rates of skin clearance) in patients with psoriasis [24,25]. A pilot study on psoriasis patients found significant improvements in both psoriasis severity and quality of life in the MBCT group compared to treatment as usual (TAU) controls [26]. Another similarly designed trial found significant decreases in levels of depression in the MBCT group compared to TAU controls [27]. Moreover, significant improvements in mindfulness levels and reduced stress levels were measured in AD patients after a standard mindfulness intervention (MBSR) [28]. A survey conducted among 120 patients found that higher levels of naturally occurring (dispositional) mindfulness are associated with lower levels of psychosocial distress in dermatology patients such as social anxiety, anxiety, depression and skin shame and higher levels of quality of life [29]. In clinical contexts and beyond, mindfulness is being employed as an instrument for behavior change, in particular for breaking automatic habits [30]. It has been suggested that this mechanism might be potentially beneficial for breaking the itch–scratch cycle [14].

However, in spite of first attempts to examine the effects of mindfulness in the context of dermatological conditions, the research area is largely untapped. Except for two small pilot studies [28,31], past work primarily focused on psoriasis, not on atopic eczema populations. Itch-related effects and mechanisms have not yet been examined. Furthermore, despite promising results about symptom-specific mindfulness protocols in psoriasis patients [25], a fully elaborated program specifically tailored to demands of skin patients is lacking. The present article reports on the effects of a newly designed Mindfulness-based Training for chronic Skin Conditions (MBTSC) intervention on an atopic eczema sample. MBTSC was developed with the goal of improving self-regulation including stress management and emotion regulation in patients and to help in coping with disease symptoms such as itch and scratching. The primary objective of this pilot study was to examine whether the MBTSC leads to a reduction in symptom severity and stress levels. The secondary

objective was to assess its impact on anxieties and depression as well as to determine the feasibility and acceptability of the intervention. This study was conducted in parallel with another pilot with AD patients in which the effects of a standard mindfulness intervention (MBSR) were measured [28]. Because of the small group sizes and the different lengths of the interventions, comparability of the two groups would have been limited. For this reason, it was decided to treat these two studies separately.

## 2. Materials and Methods

### 2.1. Participants and Study Procedures

After having obtained approval by the Ethics Committee of Technical University Munich, ten adult patients with diagnosed AD were recruited from the university dermatology clinic and from local dermatological ambulances. Patients aged between 18 and 65 were eligible to participate in the trial. A third inclusion criterion was sufficient proficiency in the German language to fill in questionnaires appropriately. Exclusion criteria were psychiatric and cognitive disorders (psychosis, personality disorders or dementia). All patients meeting inclusion criteria were previously informed about the basic objectives and contents of the intervention (e.g., that the intervention aims at improving self-perception and coping with the disease, as well as quality of life). On informed consent from the patients, questionnaires were handed out to be completed at home and returned to a physician on the first day of the intervention. Furthermore, questionnaires were handed out immediately after the intervention and after the follow-up session after 3 months.

### 2.2. Intervention

The MBTSC protocol consists of seven weekly group sessions. Participants are required to perform daily home practice including formal meditation and informal mindfulness practice. Three months after completion of the intervention, a booster session was offered to freshen up mindfulness practice and to discuss the effects of the intervention on AD and efforts to integrate mindfulness practice in daily life.

The intervention curriculum (see Table A1) is based on the standard 8 week MBSR curriculum [32] with the following alterations: (1) the program was shortened from 26 h (MBSR) to 14 h. Instead of eight weekly sessions of 2.5 h and an all-day class of practice after session six ("the retreat day," 6 h); the new protocol comprises seven 2 h sessions without the retreat day. In addition, daily home practice requirements were reduced from 45 min (MBSR) to 30 min. The more compact curriculum was developed in response to demands for a lighter contemporary intervention format that is more compatible with the busy daily agenda of participants [33]. It was suspected that, especially for chronically ill patients, the demanding time commitment of MBSR could be a barrier to participation or regular attendance because they are burdened with time-consuming skin care and therapy on a daily basis. These additional time constraints might prevent compliance with practice requirements unless the participant has a very high level of motivation to dedicate an additional daily 45 min and 2.5 h per week to exercise.

(2) Additional educational elements on disease and disease management were incorporated into the curriculum and in the reading material as well, including the following aspects: (i) information on the itch–scratch cycle, automatic scratching, causes for itch and scratching; (ii) background information regarding the skin–psyche relationship. These elements were supported by practical exercises targeting the exploration and experience of AD symptoms with a non-judgmental attitude ("skin scan;" "mindful exploration of itch"), which was specifically developed drawing on previous works of experienced meditation teachers. In addition, participants were invited to write an itch diary as a means to support and integrate daily subjective pruritus monitoring. Therefore, a modified form of the patient itch diary from the German working group for patient education in childhood AD (Arbeitsgemeinschaft Neurodermitisschulung, AGNES) [34] was included in the course material. (3) A specific focus on emotions was integrated into the curriculum, with the aim to encourage participants to recognize and physically explore the emotions as they appear,

instead of automatically ignoring or suppressing them. The second aim was to, on the one hand, sensitize participants to the relationship between (negative) emotions, physiological arousal and itch/scratching impulses, and, on the other, to inform them about the relevance of inner attitude towards emotions, in particular resistance and fighting versus acceptance and self-compassion as beneficial regulation strategies. The didactic focus on emotion regulation was introduced with the intention of influencing the emotional aspects of itch processing in response to the beneficial effects on pain experience that have been reported in pain research.

To make sure that the intervention protocol met the quality standards of existing MBIs, experts including psychologists and mindfulness teachers with experience in clinical contexts were interviewed. Furthermore, to ensure that the intervention was tailored to patient needs and requirements, informal, semi-structured narrative patient interviews prior to the development of the curriculum were carried out with a pre-pilot group of 7 AD patients at Humboldt University Berlin in order to ascertain acceptability and feasibility as well as to make improvements. This pre-pilot intervention was supervised by a psychologist with long-term experience in dermatology patient education who also coached the mindfulness trainer in patient education. Participant feedback, obtained through interviews and feedback questionnaires, helped to refine the MBTSC protocol.

The intervention was conducted by a mindfulness and insight meditation (Vipassana) teacher with long-term experience in Buddhist Theravada meditation. The trainer was instructed to record unusual occurrences as well as participant observations of itch and scratching after each session in a notebook.

### 2.3. Outcomes

Patients completed a series of validated questionnaires at baseline (T1), at post-treatment after 7 weeks (T2), and at follow-up after three months (T3). The self-report measures covered disease severity, stress, mindfulness, and psychological distress. Relevant clinical and demographic data were collected at baseline. At completion of intervention and at follow-up, patients additionally completed questionnaires on the acceptability of the intervention as well as questions on global changes.

#### 2.3.1. Disease Severity

Disease severity was determined using 3 questionnaires. The Person-Oriented SCORing Atopic Dermatitis (PO-SCORAD) [35] grades the extent, intensity and subjective symptoms of AD including itch intensity and quality of sleep. Scores range from 0 to 104, with severity strata having revised and newly set (0–27 = mild, 28–56 = moderate, and 57–104 = severe). PO-SCORAD has been shown to be a reliable instrument that significantly correlates with the Patient-Oriented Eczema Measure (POEM). The POEM [36] is a 7-item questionnaire for monitoring the subjective perception of atopic eczema severity. The POEM assesses the frequency of occurrence of symptoms (itching, dryness of skin, cracking of the skin, and sleep disturbance) in the past week. Severity threshold scores on the POEM have recently been revised and slightly adjusted (0–7 = mild, 8–19 = moderate, and 20–28 = severe). Itching was measured using the widely used Eppendorf Itch Questionnaire (EIQ) [37]. The validated multidimensional inventory measures subjective experience of itch based on 4 subscales, which in part are reported to significantly correlate with SCORAD (Spearman's correlation coefficient of 0.33). The subscales assess (i) sensory itch descriptions; (ii) affective evaluations of the itch experience. Participants rate each sensory and affective item based on a 4-point Likert scale ranging from "not true" (0) to "describes exactly" (4); (iii) itch intensity, rated based on a VAS scale; (iv) most frequent itch-relieving (pruritofensive) reaction habits to itch. The scale suggests a list of 10 descriptors including scratching, distracting, pinching, cooling or showering, which participants are required to rate based on a 4-point Likert scale that ranges from "no" (0) to "yes" (4).

### 2.3.2. Perceived Stress

In order to determine the level of subjectively perceived stress, a shortened and translated version of the Perceived Stress Questionnaire (PSQ) [38] was administered. The PSQ has been documented to be a reliable instrument in the assessment of stress in psychosomatic populations and to be sensitive to change after treatment. The 20-item measure assesses emotional as well as cognitive dimensions of experienced stress based on four factors (worries, tension, joy, and demands). Participants are asked to consider the frequency of each item (worries, fears of the future, frustration, exhaustion, sense of being out of balance and under tension, joy as well as the extent of external demands), as experienced in the past month on a 4-point Likert scale ("hardly ever" to "usually"). The scale ranges from 0 to 100, with high scores indicating a high level of subjectively perceived stress with a cut-off score for moderate to severe stress > 0.45.

### 2.3.3. Mindfulness

Mindfulness was measured with two validated inventories. The Freiburg Mindfulness Scale (FMI) [39] primarily assesses two mindfulness facets—awareness and acceptance. A short and validated version (FMI-14) of the questionnaire that has already been applied in other clinical trials was used for this study. The Mindful Attention Awareness Scale (MAAS) [40] measures the degree of awareness regarding specific occurrences in day-to-day life based on a 6-point Likert scale (6 = almost never; 1 = almost always). For both scales, high scores reflect greater mindfulness.

### 2.3.4. Anxiety and Depression

The Hospital Anxiety and Depression Scale (HADS) [41] is a widely used instrument exhibiting good reliability. Consisting of 14 items, the HADS questionnaire is divided into two subscales measuring the levels of anxiety (HADS-A) and depression (HADS-D). The scoring of each scale ranges from 0 to 28, with higher scores indicating greater levels of anxiety and depression. With a cut-off score $\geq$ 8 (8–10 = mild anxiety/depression, 11–14 = moderate anxiety/depression, and 15–21 = severe anxiety and depression), the HADS scale is reported to have excellent predictive validity for identification of approximately 70%.

### 2.3.5. Intervention Acceptability and Feasibility—Global Impressions of Change

To explore the acceptability and feasibility of the intervention, two qualitative questionnaires were administered at T2 and T3. An adapted multidimensional feedback questionnaire from Freiburg University (unpublished), which was developed for assessing mindfulness-based interventions, was used. The feedback questionnaire assesses (i) treatment satisfaction including feedback on specific themes, exercises, reading and audio material and other aspects of the intervention; (ii) subjectively experienced changes in symptoms; (iii) global changes. The questionnaire includes rating options based on Likert scales and open questions in order to gain more in-depth information on what worked and what did not work in the intervention. Furthermore, an adapted version of the widely used Patient Global Impression of Change (PGIC) scale was employed at follow-up to ascertain patient ratings of overall improvement after the intervention. The adapted version assesses (i) improvements regarding general health; (ii) changes regarding specific aspects (itch, sleep impairment, mood, stress response, quality of life, relationships, and social activities). All items are rated based on a 5-point Likert scale (much better–much worse). Feasibility was additionally determined by the degree of home-practice compliance and completion rates. In order to be considered completers, participants had to attend 3 or more intervention sessions.

### 2.4. Data Analysis

Statistical analysis was conducted with SPSS IBM statistics version 26. Except sociodemographic and acceptability variables, data were analyzed only for completers of all

three measurements in order to obtain realistic effect size estimates for a larger study. Socio-demographic variables and qualitative data were analyzed according to intention to treat to be able to use all available data. Descriptive statistics for quantitative variables were presented as the mean ± standard deviation (SD). The significance of differences between the mean scores at baseline and post-treatment was calculated using paired t-tests and Cohen's d. Due to the small sample size N = 6), the standard distribution was not given and hence it would not have been reasonable to perform any more in-depth on-going statistical analyses such as correlations. Statistical power estimations showed that given an alpha of 0.5, 27 participants would have been required for 80% power to detect a large effect size ($d = 80$), which meant that the actual recruited sample rendered fairly low statistical power with an alpha of 0.2–0.3. Qualitative data obtained from the questionnaires were analyzed based on content analysis. Raw data were collected by University Coburg, and the final analysis and presentation of data were conducted by an independent statistician.

## 3. Results

A total of 9 of 10 eligible participants presented themselves for the intervention and completed the questionnaire at baseline. In total, 6 of 9 patients completed the questionnaires at T2 and T3 (a 67% completion rate). Statistical analyses only included completers ($N = 6$). Descriptive statistics and qualitative data were analyzed based on intention to treat ($N = 9$). The majority of participants were female (78%), with moderate disease severity, and ranged in age from 27 to 73 years (see Table 1).

**Table 1.** Subject Characteristics.

| Variable | *N* | | Value |
|---|---|---|---|
| Gender | 9 | Women *n* (%) | 7 (77.8) |
| Age | 9 | Mean (SD) | 48.10 (15.00) |
| | | Min–Max | 27–73 |
| Secondary school qualification | 8 | High *n* (%) | 6 (66.7) |
| | | Low | 2 (22.2) |
| Previous meditation experience | 9 | No *n* (%) | 9 (100) |
| Experiences with similar methods | 9 | Yes *n* (%), (Methods) | 2 (22.2), (Yoga, Thai Chi) |
| | | No | 7 (77.8) |
| Mean duration of symptoms per year | 9 | Mean (SD) | 10.33 (2.7) |
| Number of in-patient treatments in the last 5 years | 9 | Mean (SD) | 2.63 (2.66) |

Note. *N* = number of patients; SD = standard deviation.

As depicted in Table 2, mean disease severity scores showed moderate decreases in subjectively perceived symptoms at post-treatment, with medium effect sizes in the case of PO-SCORAD ($p = 0.290$; $d = 0.65$).

Likewise, the itch ratings (EIQ) dropped after the intervention and again at follow-up, although the effect size was not significant ($p = 0.407$; $d = 0.41$). A noticeable difference in pre- to post-treatment scores between descriptive and affective components of itch perception can be seen in Figure 1. While the descriptive assessments only marginally decreased from T1 to T2, the affective-evaluative ratings, indicating the degree of emotional suffering as part of the itch experience, showed a much larger though not statistically significant reduction between baseline and post-intervention ($p = 0.320$; $d = 0.51$).

Subjective stress (the PSQ) and mindfulness levels (the FMI and the MAAS) improved moderately, although without significant baseline to post-treatment changes. Distress scores (HADS) showed small to large increases in effect size, with statistical significance in the case of depression ($p = 0.033$; $d = 1.43$), but not anxiety ($p = 0.530$; $d = 0.28$). Regarding self-perceived changes at post-treatment ($N = 9$), the majority of participants ($n = 7$) reported improvements in symptoms or changes in coping with the disease. Two participants

observed improvements in itch and scratching behavior, while an equal number did not. Other observed changes include (i) more awareness of itching and scratching impulses; (ii) increasing sensitivity to and understanding of symptom causes and correlations with stress and negative emotions; (iii) less emotional reactivity to and more acceptance of disease symptoms. None of the participants reported any worsening. With respect to the feasibility of the intervention, there was a 100% completion rate, where attending $\geq 4$ sessions = completer (4 participants missed none of the sessions, 5 missed 1–2). Regarding satisfaction with the intervention, all participants (*n* = 9) reported being satisfied with the intervention; eight would recommend the intervention to other patients, one with the indication that the intervention should to be on a voluntary basis and noting that mindfulness might not be helpful in acute phases of exacerbation.

**Table 2.** Means, Standard Deviations and Effect Sizes (Cohen's d).

| Measure (Scale, Range) | N | Pre (T1) M (SD) | Post (T2) M (SD) | 3 MFU (T3) M (SD) | *p* T1–T2 | T1–T3 | *d* |
|---|---|---|---|---|---|---|---|
| Eczema Severity (POEM, [a] 0–28) | 5 | 14.00 (7.48) | 13.60 (7.20) | 13.20 (7.85) | 0.882 | 0.699 | 0.07 |
| Eczema Severity (PO-SCORAD, [b] 0–103) | 4 | 28.05 (12.12) | 21.64 (9.18) | 25.56 (18.06) | 0.290 | 0.651 | 0.64 |
| Pruritus—Descriptive (EIQ, [c] 0–160) | 5 | 26.6 (18.8) | 22.8 (11.4) | 20.0 (8.0) | 0.710 | 0.558 | 0.18 |
| Pruritus—Affective (EIQ, 0–160) | 5 | 46.4 (43.3) | 24.2 (13.3) | 21.0 (14.5) | 0.320 | 0.326 | 0.51 |
| Stress (PSQ, [d] 0–100) | 6 | 43.42 (20.00) | 36.94 (24.77) | 42.51 (16.15) | 0.367 | 0.811 | 0.40 |
| Presence (FMI, [e] 1–4) | 6 | 2.50 (0.52) | 2.58 (0.65) | 2.71 (0.41) | 0.807 | 0.584 | 0.10 |
| Acceptance (FMI, 1–4) | 6 | 2.65 (0.65) | 2.67 (0.53) | 2.75 (0.35) | 0.947 | 0.722 | 0.03 |
| Mindfulness (MAAS, [f] 1–6) | 6 | 2.46 (0.45) | 2.67 (0.48) | 2.48 (0.49) | 0.172 | 0.771 | 0.65 |
| Anxiety (HADS-A, [g] 0–28) | 6 | 8.67 (1.63) | 9.00 (1.67) | 9.33 (2.25) | 0.530 | 0.328 | 0.28 |
| Depression (HADS-D, 0–28) | 4 | 11.25 (0.96) | 12.50 (0.58) | 12.00 (1.41) | **0.033** | 0.477 | 1.43 |

Note. Pre/Post = pre-/post-treatment; *N* = number of patients; 3 MFU = 3 month follow-up; effect size (Cohen's d) including 95% Cis; bold letters indicate statistical significance. [a] Patient-Oriented Eczema Measure. [b] Person-Oriented SCORing Atopic Dermatitis. [c] Eppendorf Itch Questionnaire. [d] Perceived Stress Questionnaire. [e] Freiburg Mindfulness Scale. [f] Mindful Attention Awareness Scale. [g] Hospital Anxiety and Depression Scale.

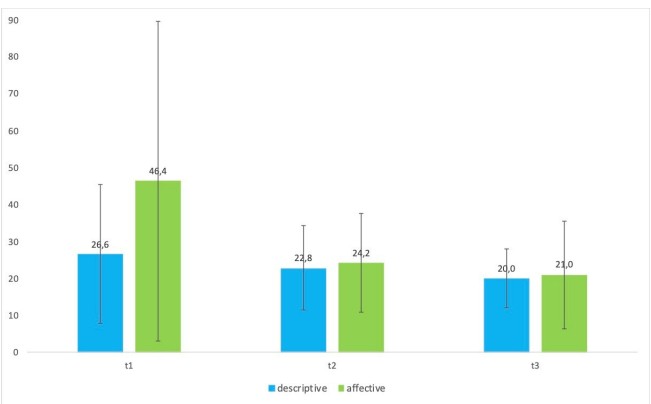

**Figure 1.** Differences in descriptive and affective itch experiences at baseline, post-treatment and follow-up (mean sum scores and standard deviation).

## 4. Discussion and Conclusions

To the author's knowledge, this is the first study to examine the effects of a mindfulness-based intervention with psychoeducational elements in AD conditions and pruritus, and suggests that mindfulness might have a promising role in reducing itch and symptom severity. The small improvements in self-reported eczema severity (the POEM and PO-SCORAD) at post-intervention are in line with previous results on psoriasis patients [25,26]. It is difficult to determine which intervention component (mindfulness facets, psychoeducational elements, etc.) resulted in the positive effects since there was no control group. More research needs to be conducted to ascertain whether the adapted intervention with psychoeducational elements has additional benefit compared to a standard mindfulness protocol (MBSR).

The decrease in subjective itch perception at post-intervention and at follow-up points to a potential modulating effect of mindfulness on itch. These effects of mindfulness are compatible with well-documented effects on pain reduction in chronic pain patients [42], but have never been observed in chronic itch conditions. Furthermore, present data seem to confirm a suggested potential mechanism of change regarding MBIs that has been detected in pain conditions. Furthermore, present data seem to confirm a suggested potential mechanism of change regarding MBIs that has been detected in pain conditions. Studies found that while mindfulness-training does not primarily decrease pain intensity, it affects the pain ap-praisal attenuating emotional distortions of painful stimuli perception by encouraging nonevaluative contact with the world [43]. These findings and explanations align well with obtained data regarding the distinctly larger decrease in affective itch values compared to descriptive sensory itch values. However, itch in dermatologic conditions is a very complex mechanism and the small sample size prevents in-depth on-going evaluations. Further research is needed to individuate the potentially itch-modulating factors and to understand the mechanisms of change of the intervention. Future studies with larger samples will hopefully help elucidate why these observed tendencies in itch reduction did not show in all participants. In fact, as qualitative study data about itch perception indicate, 30% of the participants reported a change in itch perception, while others did not. Many factors may account for this heterogeneity of results, such as the impact of physiological itch- mediators (e.g., dryness of skin), hence it is difficult to say more on this.

Against expectations, self-reported itch-relieving strategies (EJF) did not seem to be affected by the intervention. MBIs are known for particularly emphasizing the development of a benevolent and caring attitude towards oneself, which can result in (a) mindful awareness and assessment of one's internal needs and external demands and (b) intentional engagement in specific practices of self-care to address needs and demands in a manner that serves one's well-being and personal effectiveness [44]. Based on these observations, it was expected that an increase in self-compassion might translate into a change in pruritofensive patterns, away from skin-damaging strategies such as "scratching until the skin bleeds", towards more skin-protective scratching alternatives such as cooling or stroking. According to this hypothesis, a change would be reflected in the sum scores of respective items, which was not the case. Instead, data seemed to suggest a relative stability of reaction patterns, which might give way to a different conclusion. Firstly, itch can be such an intrusive experience and scratching such a primordial, deeply ingrained reaction that a 7 week intervention might not be sufficient to change deeply rooted behavior patterns. Additionally, scratching might be associated more with itch intensity then with psychological factors, such as subjective stress, which seems to be supported by study data showing fairly moderate changes in itch intensity. This explanation is in accord with participant qualitative feedback that with subtle itch it was easier to maintain an attitude of mindful non-reactivity and to control scratch-impulses than with strong itch.

The moderate improvements in subjectively perceived stress at post-treatment are in alignment with the widely confirmed findings of mindfulness alleviating stress response [17]. However, the improvements observed are rather marginally compared to hitherto measured significant effects on stress. Likewise, the elevated stress levels at T3

are inconsistent with previous findings documenting largely stable and lower stress levels for post-treatment and follow-up measurements, which is taken as a proof of long-term effects of MBIs [45]. An explanation for this might be the decreased amount of mindfulness practice in the new condensed intervention, which may have diluted the stress-reducing effects. However, other factors such as sample size might also account for small effect sizes, which calls for further evidence-based examination.

The somewhat surprising increases in anxiety and depression ratings with significant effect size in the case of depression are the opposite of what was hypothesized and counter to previous literature [26]. Another finding which appears contradictory to current research relates to the association between anxiety and itch perception. According to prevailing constructs asserting a mutually reinforcing relationship between itch and anxiety [46], increasing levels of anxiety and depression should correlate with increasing itch perception, or vice versa, decreasing levels of itch attenuating anxiety and depression, which was not the case. It is difficult to provide reliable explanations at this early stage of research, in particular because only 4 of 6 participants answered the depression questionnaire in full, hence the correlation between itch and anxiety or depression might be pure coincidence. Further, the pathways of interaction between itch and emotions are unknown as of yet. Nevertheless, these results allow for preliminary conclusions that might be helpful regarding the development of future psychological and psychoeducational patient interventions addressing emotion regulation. The mechanisms of change of the MBIs in emotion regulation have not yet been clearly understood, but it has been suggested that MBIs might affect emotions via different strategies and pathways of action, including cognitive reappraisal, decentering ("top-down approach") and acceptance ("bottom-up approach") [47,48]. A review of mindfulness research and comparable MBI studies with dermatology patients with depression and anxiety reveals that the intervention used was Mindfulness-based Cognitive Therapy (MBCT) [26,27]. MBCT is an adapted MBI, specifically developed for dealing with depression symptoms. MBCT interventions primarily emphasize a top-down strategy to modulate affective response in focusing on dysfunctional cognitive processes, such as ruminative and negative thoughts. In light of these findings, it might be reasonable to assume that the specific set of emotion regulation strategies employed by the newly developed MBTSC protocol, emphasizing the experiential exploration of feelings (bottom-up), might have resulted in temporarily increased perception of anxiety and depression. At the same time, the invitation to feel and accept negative feelings might have helped to decrease vegetative hyperarousal and decrease itch. As such, it may be cautiously concluded that the development of psychological patient interventions and future research into their effects needs to take into account that different strategies for emotion regulation can, on the one hand, have different effects on symptoms and, on the other hand, can influence the disease process in chronic skin conditions via different biopsychosocial pathways and processes.

Altogether, this study indicates that participation in an MBI as adjunct therapy to standard treatment for AD might be beneficial. Although preliminary, the study data about the effects on mindfulness add to present theory building in helping to illuminate the still poorly understood mechanisms of change in the mindfulness–AD/itch relationship. First, it provides preliminary evidence that mindfulness might have a modulating function in itch perception, in particular regarding the affective dimension of itch experiences. Secondly, this study highlights a new pathway of change. Beyond modulating the stress–AD/itch axis, as generally hypothesized, mindfulness might have direct effects on itch by modulating the cognitive processes of itch perception.

Because this was a pilot study, it inevitably has its limitations and the results should be appreciated as preliminary and interpreted with caution in several regards. Since there was no control group acting as reference, the results could have been interpreted erroneously, and therefore should be considered as having limited validity. Furthermore, the uncontrolled study design restricted the possibility of isolating the specific effects of the intervention, excluding other potential unspecific aspects of the intervention. Finally, it cannot be ruled out that the expectations of the participants and placebo-related processes

may account for the effects of the intervention. The modest sample size reduced power to identify significant results and limits the ability to generalize the results. Moreover, this study relied primarily on self-reported data and hence the results are liable to bias. Due to a relatively short follow-up period, it was not possible to assess the long-term effects of mindfulness. Ultimately, the measures employed to assess mindfulness questionnaires, though validated, have been variously criticized as inadequately measuring mindfulness skills [49,50]. Large-scale replications are needed to allow for generalizability of the results. Similarly future trials need to employ active control groups to isolate the effects of different components of the intervention including mindfulness skills and psychoeducation.

**Funding:** This research received no external funding.

**Institutional Review Board Statement:** Ethikkommission, Fakultät für Medizin, Technische Universität München, Germany (Project number 14/14, approved 13 February 2014).

**Informed Consent Statement:** Informed consent was obtained from all subjects involved in the study.

**Data Availability Statement:** Not applicable.

**Acknowledgments:** The author thanks Christina Schut and Ulf Darsow for their helpful comments on the manuscript.

**Conflicts of Interest:** The author declares no conflict of interest.

## Appendix A

**Table A1.** Structure and Content of the MBTSC Protocol.

| Session | Content, Mindfulness Exercises and Skills | Psychoeducational Elements |
|---|---|---|
| I. | <ul><li>Introduction to the intervention</li><li>Mindful breathing exercise</li><li>Basic principles of mindfulness practice: presence, non-judgmental and open attitude, acceptance, kindness towards oneself</li><li>Handout and homework: [1] mindful breathing</li></ul> | <ul><li>Inquiry and discussion about the potential of mindfulness for skin issues: How can mindfulness help?</li><li>Key message: Eczema and itch as a valuable source of information</li></ul> |
| II. | <ul><li>The nature of the mind: autopilot, mind wandering, reactivity; conscious response instead of automatic reactions</li><li>Mindful breathing exercise; combined body and skin scan</li><li>Handout and homework: breathing meditation combined with body awareness</li></ul> | <ul><li>Discussion on the role of one's attitude towards AE; option of changing our relationship to symptoms and seeing AE and itching as valuable signalers</li><li>Reading material: mindfulness with AE and itch. How does mindfulness help and how to use symptoms to develop awareness and right attitude?</li><li>Homework assignment: consciously register moments of automatic scratching and other reaction habits to itch</li></ul> |
| III. | <ul><li>Three kinds of sensations and mental reaction patterns (grasping, aversion, ignoring)</li><li>Aspects of mindful attitude (letting go of expectations, not judging, not grasping, patience, kindness)</li><li>Bodyscan</li><li>Handout and homework: bodyscan</li></ul> | <ul><li>Inquiry about itch experience; information: there are different types and qualities of itch that may give information about potential causes and correlations (sweat, dryness of skin, emotions).</li><li>Introduction itch diary [2]</li><li>Homework assignment: consciously register itch whenever it; check emotional, physical states and thoughts that accompany itch experience including emotional reactivity to itch.</li></ul> |

**Table A1.** *Cont.*

| Session | Content, Mindfulness Exercises and Skills | Psychoeducational Elements |
|---------|-------------------------------------------|----------------------------|
| IV. | • Overcoming challenges and difficulties in meditation and in everyday life by developing wholesome qualities of mind (7 factors)<br>• Breathing meditation, walking meditation<br>• Handout and homework: walking meditation | • Handout: reading material on the skin–psyche relationship [3] |
| V. | • Dealing with (negative) emotions and resistance; emotion regulation strategies (suppression, fighting, ignoring); mindful approach to emotions<br>• Loving-kindness meditation (Metta)<br>• Handout and homework: Metta-meditation | • Discussion about the relationship emotions, dysfunctional regulation and itch/AE exacerbations; relevance of appropriate emotion regulation strategies for AE<br>• Reading material: mindfulness with emotions (emotional awareness, physical exploration, labeling; opening up to and letting go of inner resistance to emotion) |
| VI. | • Mindful (self-)compassion: acceptance and friendliness towards oneself<br>• Breath meditation and mindful movement exercise<br>• Handout and homework: developing a practice for yourself individual choice of practice (bodyscan, Metta, walking meditation) | • Suggestion: self-compassion as helpful coping strategy to attenuate inner tension, worries and anxiety in phases of AE exacerbation |
| VII. | • Reflection about lessons learned<br>• Outlook: how to integrate mindfulness in everyday life<br>• Mindful movement, mindful observation of thoughts exercise | |

[1] The material included guided audio-meditations and participants were asked to practice for at least 20 min daily either formally or informally. [2] A modified form of the patient itch diary from the German working group for patient education in AD (Arbeitsgemeinschaft Neurodermitisschulung, AGNES) was used. [3] Excerpts were used from Detig-Kohler C. Hautnah: im psychoanalytischen Dialog mit Hautkranken: Psychosozial-Verlag 2002.

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
