# Peer review of "An Open Trial on the Feasibility and Efficacy of a Mindfulness-Based Intervention with Psychoeducational Elements on Atopic Eczema and Chronic Itch"

_psych, doi:10.3390/psych4020014_

Round 1
Reviewer 1 Report
It is a needed search in psychodermatology and very well designed study.
However, I would suggest a title as " an open trial on efficacy of mindfullness interventions on chronic itch in atopic dermatitis "
On the other hand, in overall evaluation, mindfullness intervention did not have an expected effect on itching . It would be fine to examine physical components of itch in AD such as xerosis, sweating or adding that possibility to discussion. Because, this technic address directly emotional comonent of itch.
Author Response
Point 1: I would suggest a title as "an open trial on efficacy of mindfullness interventions on chronic itch in atopic dermatitis "
Response 1: I'm not quite sure on what grounds you suggested a title change. I assume the title should read more smoothly. I'm happy to accept that suggestion, but I've reworded your version a bit. The wording "on the efficacy of mindfulness on chronic itch in atopic eczema" would mean to me that only chronic itch was studied as a variable, which is not true since other outcome parameters were implied, such as disease severity, stress and psychological distress. Therefore, I would leave it at "and", if you agree. Even though there is no reason to mention itch before the other variables in the title, which is perhaps what you were aiming at with the title suggestion, the mention of "itch" is important to me for the following reason. It is a distinguishing feature for the study and sets it apart from similar studies on the topic. I am open to further suggestions.
Point 2: In overall evaluation, mindfullness intervention did not have an expected effect on itching . It would be fine to examine physical components of itch in AD such as xerosis, sweating or adding that possibility to discussion. Because, this technic address directly emotional component of itch.
Response 2: I very much appreciate your suggestion to discuss potentially biasing influence of other moderators of itch perception on the collected data. I have included your suggestion in the discussion section.
Regarding your conclusion on the effects of the intervention on itch, I'm afraid I can't quite agree with you. The study showed a tendency to improve itching, even if not significantly. Since it was a pure feasibility study, we were not interested in significant results, but only to see if and in which areas the training had an effect.
It is a very good suggestion to specify on physical components on itch, thank you! I have included it into the article in the discussion section.

Reviewer 2 Report
The study is interesting and has potential. I do have some major comments:
Abstract: Misleading. You mention 10 patients with AD (please write all abbreviations in full when given for the first time). Not all 10 recruited patients were included.
Introduction: You do not mention studies on AD treated with standard minfulness. If you will perform a more compact course you should mention how a standard course helps. Is there research on this? In the next sentences you write that there are no existing studies on AD.
Minor: 'strong' itch. This is not always true
TAU: Please write in full først time used.
Materials and Methods:
Please mention what results are obtained for adult AD patients with the standard protocol. More compact protocol than standard but based on what grounds? Mention the effect of standard course on AD patients. If standard protocol is never used previously, how can you assume that a compact protocol will reflect the reality? Did the pilot use a standard or a compact course?
The patient-itch diary is originally not designed to be used in adults. Can you explain the validity of a modified (not tested out) diary? Is the Eppendorf itch scale the one you mentioned previously, that is used for children? This is somewhat unclear.
Is the FMI-14 validated? It may not be validated even if used in clinical trials previously. Please mention.
You mention that you used a questionnaire developed for assessing mindfulness-based interventions for chronic pain. Can it be used with the same validity for assessing AD-related symptoms?
You excluded patients with less than 3 interventions. Are patients with only 3 (or 4) interventions enough to conclude anything? Do you mean that you had 10 patient, 6 of them completed and completion was registered if at least 3 of X? sessions were attended? Please explain more precisely. Exactly how many patients attended all of the planned interventions? Why did you not include 27 patients if the necessary minimal number was calculated to be 27? Actually, having such low number of patients completing all sessions could rather be reason to call this a pilot study, with limitations on conclusions. You call it a study. Please modify.
Table 3: Is N number of patients with completed questionnaires?
You should describe your results as 'indication of', 'suggests' or 'trends', but cannot conclude anything with certainty. If 27 are needed and only 4-6 have actually participated and given valid answers to all questions the results cannot be interpreted.
Discussion and conclusion: You cite references 19 and 26 as 'in line with previous results' on eczema severity (POEM and PO-SCORAD), but those srudies are not performed on AD patients. Please specify what you mean.
How significant and valid are the results in your data to conclude with 'distinctly larger decrease in affective itch' (page 8 Line 322 and 323).
Page 8, Line 333: This may solely be because of the low number of participants. Cannot conclude anything.
Page 9, Line 358 and 359 - you call it a potential explanation, but I think you could call it a realistic explanation, inconclusive results because of low sample size and modified MBCTS.
The contradictory results on itch and depression/anxiety may solely be caused by low sample size. Only 4 patients completed HADS-D. The results may be pure coincidence and therefore not surprising.
Line 371 and 372 - 'results allow for important conclusions' - I am not certain that this can be claimed for your study. The conclusion may be that more studies with higher number of patients on AD are necessary.
A summarized conclusion is missing.
Other:
You use sigle person in the discussion. Is the paper written by only one author and the study performed with no collaboration from others?
Please give limitations in more detail. You do not mention selection bias: only those interested and believing in MBTSC might wish to participate. Can not conclude that applicable for other patients (less motivated for MBTSC). Participation bias should also be mentioned. Feeling that minfulness is helpful or not will have an effect on wanting to participate or not, and placebo efefct can only be determined if there was a controll group.
Small number of patients. Misleading as 10 patients. Please give the true number in abstract and in results.
Bias: Having to attend regular sessions which are time consuming may for some be an extra stress factor. If the MBCTS is perceived as burdensome, but necessary it may rather be perceived as a burden. This should be discussed.
Author Response
See document
